# Bias Spillover in Language Models: A Review of Political Alignment, Regional Fragility, and Multi-Axis Risks

## Abstract

Efforts to mitigate social bias in large language models (LLMs) often target dimensions such as gender or political ideology in isolation. Yet interventions along one axis frequently propagate to others, a phenomenon we term *bias spillover*. This paper reviews over 80 studies, synthesizing empirical and theoretical evidence of cross-axis interference in model behavior. We define bias spillover as the unintended alteration of behavior on one social axis when mitigating another, driven by representational entanglement, competing fine-tuning objectives, and structural fairness trade-offs. These effects align with well-known optimization pathologies such as Goodhart's Law, reward hacking, task interference, and impossibility results in algorithmic fairness, highlighting spillover as a fundamental, not incidental, challenge. We document observed spillover cases, for instance, political fine-tuning shifting emotional tone and moral framing, or gender balancing distorting age distributions and identify blind spots in current audits, including poor coverage of multi-axis and non-Western contexts. We conclude by introducing a typology of auditing frameworks and recommending mitigation strategies that explicitly account for entangled social representations, moving beyond isolated fairness metrics toward spillover-aware evaluation of LLMs.

## 1 Introduction

The rise and rapid advancement of large language models (LLMs) has fundamentally changed language technologies (Brown et al., 2020; Devlin et al., 2019; Raffel et al., 2020). With the ability to generate human-like text, as well as adapt to a wide array of natural language processing (NLP) tasks, the impressive capabilities of these models have initiated a paradigm shift in the development of language technologies. Rather than developing task-specific models trained on moderately sized datasets, contemporary research and practice increasingly leverage large language models (LLMs) as foundation models, adapting them via fine-tuning or prompting to address a wide range of downstream tasks (Bommasani et al., 2021). Even without fine-tuning, foundation models now offer few- or zero-shot capabilities across a wide range of tasks such as classification, question-answering, reasoning, and information extraction (Brown et al., 2020; Kojima et al., 2022; Liu et al., 2023; Radford et al., 2019; Wei et al., 2022; Zhao et al., 2021).

Lurking beneath these technological successes is the persistent risk of social harm. LLMs are typically trained on massive, largely uncurated Internet-scale datasets, inheriting and reproducing stereotypes, ideological distortions, and exclusionary language that disproportionately affect marginalized groups (Bender et al., 2021; Dodge et al., 2021; Sheng et al., 2021). These harms often reflect what is broadly referred to as *social bias* - the systematic disparity in model behavior towards different demographic or ideological groups, arising from historical and structural asymmetries (Benjamin, 2020; Blodgett et al., 2017; Hutchinson et al., 2020; Mozafari et al., 2020; Sap et al., 2019; Sheng et al., 2019).

Prior work has documented the presence of such biases along many social axes, most notably gender (Blodgett et al., 2017) and political ideology (Liu et al., 2022). However, these two domains are often studied in isolation, each with their own datasets, taxonomies, and evaluation protocols. Gender bias is typically measured through occupational associations or pronoun resolution, whereas political bias is often evaluated via alignment with ideological statements or framing of issues. As a result, our understanding of how these

biases might interact either during model training or evaluation, remains underdeveloped. While several comprehensive reviews of social bias in large language models (LLMs) already exist e.g., Gallegos et al. (2024a), our goal is narrower and more focused: we aim to surface and characterize the phenomenon of *bias spillover*—how biases along different social dimensions, such as gender, race, religion, or political ideology, interact and influence one another in ways that are often unintentional and under reported.

Our central focus is the concept of *bias spillover* – the phenomenon where an intervention to mitigate bias on one axis (e.g., political ideology) inadvertently shifts model behavior along another axis (e.g., gender or race). This cross-axis side effect is not entirely novel; it echoes broader phenomena in machine learning and AI alignment. For instance, optimizing a model for a single proxy objective often leads to unexpected regressions on others, as encapsulated by Goodhart's Law (when "a measure becomes a target, it ceases to be a good measure" (Karwowski et al., 2023)) and observed in *reward hacking* scenarios (agents gaming their reward function in unintended ways (Karwowski et al., 2023)). In the context of bias mitigation, focusing on one fairness criterion can induce *adaptive overfitting* (Dwork et al., 2015)to that criterion, meaning improvements on the targeted attribute may come at the expense of performance on others. In reinforcement learning parlance, aligning a model to particular preferences can incur an *alignment tax* i.e. a cost in performance on unrelated tasks (Ouyang et al., 2022). Similarly, in multi-task NLP systems, one task's optimization can interfere with others and sequential fine-tuning can cause *catastrophic forgetting* of previously learned behavior (Goodfellow et al., 2015).

Bias spillover can thus be viewed as a domain-specific instance of these general trade-offs: a fairness intervention on one axis may unintentionally create or worsen biases on another. This matters because fairness strategies that ignore such inter dependencies risk reinforcing or even creating new harms. For example, political fine-tuning has been shown to alter a model's emotional tone and moral framing, while adding gender-balancing constraints in a generative model distorted the age distribution of its outputs (He et al., 2023), (Shukla et al., 2025). In this review, we analyze recent methods and benchmarks that reveal bias spillover effects (whether by design or incidentally), highlight key gaps in current auditing practices, and argue for more interaction-aware evaluation frameworks. Rather than offering a broad taxonomy, our aim is to surface the methodological blind spots that prevent today's tools from capturing spillover and intersectional risks in LLM behavior. In this work, we have placed special emphasis on political bias, because it operates as a high-order axis that tends to influence other social bias dimensions such as gender, religion, region, and class, through framing, emotional inflection, and moral tone. Political fine-tuning can reshape a model's broader worldview in ways that cascade into those other axes. Indeed, recent structure-level audits show how interventions targeting one axis often ripple across others, reinforcing the need for intersectional and multi-axis auditing paradigms (Weidinger et al., 2022).

## 2 Political Bias in LLMs

### 2.1 Definitions and Typologies of Political Ideology

Political bias is typically defined with respect to specific ideological axes. The most common is the unidimensional *left–right* spectrum, where "left" often connotes progressive or egalitarian positions, and "right" typically denotes conservative or hierarchical orientations Feng et al. (2023). However, this one-dimensional framing can oversimplify complex political views, especially on issues like state intervention, individual freedoms, or identity politics. Multidimensional typologies such as the *Political Compass* introduce a second axis, often labeled as *libertarian–authoritarian*, to capture the interplay between economic and social ideologies Feng et al. (2023). More recent approaches extend this to data-driven ideological spaces derived from political text corpora or survey embeddings Röttger et al. (2024).

### 2.2 Methods for Political Bias Auditing

Political bias auditing in large language models (LLMs) includes both behavioral evaluation and architectural intervention. Table 1 outlines major strategies, ranging from direct testing to fine-tuning. These approaches differ in terms of introspection depth, bias assumptions, and robustness to model evasion. Direct testing methods apply standardized political alignment quizzes (e.g., Political Compass) to place models along

ideological axes. While these consistently reveal social left-leaning tendencies in commercial LLMs, they suffer from calibration flaws, oversimplified spectra, and constrained response formats (Rottger et al., 2024). Indirect and task-based methods like PRISM (Azzopardi & Moshfeghi, 2024) instead use implicit cues in generative prompts to uncover latent ideological stances, providing better resistance to model evasion.

User perception studies, where annotators rate the political slant of model outputs, often confirm perceived left-leaning biases even among left-leaning raters (Rottger et al., 2024). Content and style analysis decomposes outputs into thematic and rhetorical dimensions to detect subtle framing patterns (Bang et al., 2024), while target-oriented sentiment classification substitutes political names into fixed prompts to reveal sentiment asymmetries (Liu et al., 2021b). Experimental manipulation through fine-tuning on partisan corpora allows for direct ideological steering. As shown in Table 9, parameter-efficient fine-tuning (PEFT) methods such as LoRA, QLoRA, and Direct Preference Optimization (DPO) have been applied to align models like LLaMA-2/3 and Mistral with curated ideological corpora (Agiza et al., 2024; Stammbach et al., 2024; Chalkidis & Brandl, 2024).

However, political bias auditing remains methodologically fragmented, with significant variance across dataset design (e.g., source corpus, party-labeling granularity), alignment objectives (e.g., stance conditioning vs. preference modeling), and evaluation metrics (e.g., sentiment shift, moral tone, factuality). Among these, the absence of shared datasets and benchmarking protocols presents the most serious barrier to generalizability. Variability in PEFT methods complicates reproducibility, but the inconsistency in evaluation pipelines including incompatible taxonomies and metrics makes cross-study comparisons especially difficult.

Table 1: Overview of prominent methodological approaches for auditing political bias in large language models (LLMs). Each method captures different facets of political alignment—from explicit test-based assessments to indirect behavioral probes and experimental interventions. The table summarizes their core procedures, strengths, and known limitations, highlighting how these approaches collectively inform our understanding of ideological tendencies in LLMs.

| Method | Description and key features |
|---|---|
| **Direct testing approaches** | Administers standardized political orientation tests (e.g., Political Compass Test, Political Spectrum Quiz). Places models on ideological axes (economic/social). Consistently finds left-leaning tendencies in commercial LLMs on social issues. |
| **Indirect and task-based approaches** | Uses techniques like PRISM (Preference Revelation through Indirect Stimulus Methodology), where models generate essays or content under assigned roles or prompts. Reveals latent ideological stances without explicit questioning. More robust against refusal or evasion (Azzopardi & Moshfeghi, 2024). |
| **User perception studies** | Human raters evaluate the political slant of LLM responses to politically charged questions. Focuses on perceived bias over internal representations. Studies consistently show LLMs are perceived as left-leaning, including by left-leaning annotators (Rottger et al., 2024). |
| **Content and style analysis** | Decomposes bias into *content* (what is said) and *style* (how it's said). Analyzes emphasis, rhetorical framing, tone, and lexical choices to uncover subtle and structural political alignment Bang et al. (2024). |
| **Target-oriented sentiment classification** | Inserts names of left- and right-leaning political figures into identical sentences and measures sentiment polarity. Highlights differential treatment across political identities (Liu et al., 2021b). |
| **Experimental manipulation approaches** | Fine-tunes models on politically biased corpora (e.g., left/right news). Assesses changes in alignment post-intervention. PoliTune is a representative framework for systematic tuning and measurement (Agiza et al., 2024). |

However, the field remains technically fragmented. There is no consistent protocol regarding the choice of alignment objectives (e.g., stance prediction vs. preference modeling), fine-tuning techniques (e.g., DPO

vs. supervised instruction tuning), or evaluation setups. For instance, Stammbach et al. (2024) evaluate supervised fine-tuning (SFT), Direct Preference Optimization (DPO), and Monolithic Preference Optimization (ORPO), finding ORPO to yield the most diverse and human-aligned generations in a Swiss political context, while DPO underperforms without additional tuning (Table 9). In contrast, Chalkidis & Brandl (2024) employ only SFT with LoRA to adapt models to European Parliament party ideologies and report effective alignment particularly for ideologically consistent parties, suggesting SFT alone may suffice in some settings. These mixed results imply that no single PEFT method is consistently preferred across political alignment tasks. A broader snapshot of this methodological heterogeneity is shown in Table 9, which compares datasets, model sizes, and alignment techniques across recent studies. Overall, the diversity in model scales, data sources, and annotation schemes makes it difficult to draw generalizable conclusions.

## 3 Political Framing in LLMs: The case of US and EU

### 3.1 Differences between US and EU in political labeling and ideological structure

The political landscape in the United States is predominantly characterized by a binary party system, composed mainly of the Democratic and Republican parties. This structure encourages a relatively linear ideological framework most commonly framed as liberal versus conservative which simplifies political alignment and audit design for large language models (LLMs). In contrast, the European Union (EU) encompasses a far more complex and multipolar political spectrum. Political representation in the EU is structured around multiple *euro-parties* (transnational political groups in the European Parliament), such as the European People's Party (EPP), the Progressive Alliance of Socialists and Democrats (S&D), the Greens–European Free Alliance (Greens/EFA), and The Left in the European Parliament – Nordic Green Left (GUE/NGL) (Chalkidis & Brandl, 2024). These parties differ not just along the socio-economic left–right axis but also across other ideological dimensions including environmentalism, civil liberties, and attitudes toward EU integration (ranging from pro-EU to Eurosceptic and anti-EU). As a result, LLMs pretrained on US-centric corpora often fail to capture the ideological diversity present in EU contexts. Moreover, while US political parties tend to be more ideologically cohesive, EU parties particularly large coalitions like the EPP and S&D are often "big tents" encompassing a wide range of internal viewpoints (Stammbach et al., 2024; Chalkidis & Brandl, 2024). Table 2 summarizes the differences. This heterogeneity poses significant challenges for political bias audits and alignment in LLMs. Region specific fine-tuning, such as adapting models on European parliamentary speeches, becomes essential to accurately reflect the EU's pluralistic political environment. Without such adaptations, LLMs like ChatGPT and LLaMA-based models have been shown to default toward liberal or progressive narratives that align more closely with left-leaning euro-parties, such as Greens/EFA and S&D, thereby missing the ideological nuances of the broader European political spectrum (Exler et al., 2025; Chalkidis & Brandl, 2024).

### 3.2 Challenges in Transferring Bias Audits Across Regions

Bias auditing methods developed for U.S.-centric political contexts (e.g., Feng et al. (2023)) often do not generalize well to multilingual and ideologically complex regions such as the European Union (EU). Large language models (LLMs), including ChatGPT and LLaMA variants, are typically trained on English-language corpora that reflect American socio-political norms, leading to poor performance on region-specific political tasks. Chalkidis & Brandl (2024) demonstrate that instruction-finetuned LLMs refuse to answer prompts from the EUANDI questionnaire which is a political alignment tool for EU citizens due to their alignment with default safety and neutrality policies. To elicit responses, users must "jailbreak" the models i.e., modify the prompt phrasing in ways that circumvent built-in refusal mechanisms and enable the model to take a stance on politically sensitive issues. Even after such intervention, the models tend to favor ideological positions associated with Greens/EFA or S&D, while under representing others such as EPP or ID, revealing persistent alignment biases. Similarly, Stammbach et al. (2024) show that ChatGPT generates nearly identical liberal responses for Swiss parties across the political spectrum, ignoring key distinctions. These findings illustrate the risks of applying binary U.S.-style audit frameworks to the EU's multiparty, multilingual context. Additionally, Feng et al. (2023) show that political biases embedded in pretraining data propagate

Table 2: Key structural and ideological differences in political framing between the United States and the European Union, highlighting why political bias audits for large language models (LLMs) cannot be directly transferred between regions. The table summarizes variations in party systems, ideological dimensions, observed LLM leanings, and adaptation requirements, illustrating the need for region-specific approaches to political bias evaluation.

| Key differences | US vs. EU comparison |
|---|---|
| **Political system structure** | **US:** Binary party system (Democrats vs. Republicans). 
 **EU:** Multiparty system with coalition-based euro-parties across many ideological axes (Chalkidis & Brandl, 2024). |
| **Ideological dimensions** | **US:** Primarily single-axis (liberal vs. conservative). 
 **EU:** Multidimensional: economic (left–right), civil liberties (liberal–authoritarian), EU integration (pro- vs. anti-EU) (Chalkidis & Brandl, 2024). |
| **LLM bias observations** | **US:** ChatGPT and similar models lean liberal/progressive (Stammbach et al., 2024; Feng et al., 2023). 
 **EU:** LLMs align more with GREENS/EFA and S&D positions unless specifically adapted (Chalkidis & Brandl, 2024). |
| **Party cohesion** | **US:** Parties are generally more internally cohesive. 
 **EU:** Major euro-parties (e.g., EPP, S&D) are "big tents" with wide internal ideological range (Chalkidis & Brandl, 2024). |
| **Audit and adaptation needs** | **US:** Bias audits based on US data and spectrum are directly applicable. 
 **EU:** Requires contextual and region-specific fine-tuning to reflect political diversity (Chalkidis & Brandl, 2024). |

through to downstream tasks, reinforcing polarization and fairness gaps. Altogether, this underscores the necessity for localized, culturally aware bias audits and targeted fine-tuning when deploying LLMs beyond the U.S. context.

# 4 Understanding Spillover Bias in LLMs

## 4.1 Empirical Observations, Definitions, and Theoretical Placement of Bias Spillover

We first outline the empirical regularities observed in bias spillover and provide working definitions of the phenomenon. We then situate these observations within broader theoretical frameworks to clarify how bias spillover emerges as a byproduct of optimization dynamics in large language models. Most existing approaches to understanding bias in LLMs treat social dimensions such as race, gender, religion, or political ideology in isolation. Yet interventions or shifts along one axis frequently propagate to others, a phenomenon we term *bias spillover*. Rather than a wholly novel effect, bias spillover reflects broader trade-offs well-documented in machine learning and AI alignment research. Optimizing a model for a single proxy objective often leads to regressions elsewhere, consistent with *Goodhart's Law* and observed in *reward hacking* scenarios, where optimization exploits imperfect objectives (Karwowski et al., 2023). Similar trade-offs appear as *adaptive overfitting* to specific fairness criteria (Dwork et al., 2015), or an *alignment tax* when preference alignment harms unrelated capabilities (Ouyang et al., 2022). In multi-task and sequential learning, analogous effects manifest as task interference and *catastrophic forgetting* (Goodfellow et al., 2015).

Such trade-offs are further complicated by fundamental impossibility results in algorithmic fairness. As demonstrated by Kleinberg et al. (2016), multiple mathematical definitions of fairness—such as calibration, equalized odds, and demographic parity—cannot be simultaneously satisfied except in trivial cases. This creates inherent tensions where optimizing for one fairness criterion necessarily compromises others, a phenomenon that extends beyond single-attribute contexts to multi-axis scenarios. The impossibility of achieving all fairness definitions simultaneously provides theoretical grounding for why bias spillover effects

are not merely implementation challenges, but reflect deeper structural limitations in fairness optimization (Friedler et al., 2021). In multi-dimensional social contexts, these mathematical constraints become even more pronounced, as interventions must navigate not only competing fairness definitions within a single attribute (e.g., gender), but also across multiple intersecting attributes (e.g., gender, race, political ideology).

In the context of social bias, these trade-offs mean that attempts to address one axis of disparity can unintentionally create or worsen harms along another. For example, political fine-tuning has been shown to alter a model's emotional tone and moral framing, while gender-balancing constraints in a generative model distorted the age distribution of outputs (He et al., 2023; Shukla et al., 2025). This review synthesizes evidence of such cross-axis interactions, identifies methodological blind spots in current auditing practices, and calls for more interaction-aware frameworks to evaluate fairness in LLMs.

We define bias spillover as a phenomenon where mitigating bias along one social axis (e.g., political ideology) unintentionally alters model behavior on another (e.g., gender or race). Table 3 summarizes four mechanisms that contribute to this effect. First, during pretraining, LLMs encode socially distinct attributes in entangled subspaces, making isolated modification difficult. Marjieh et al. (2025) show a similar representational overlap for numeric and symbolic inputs. Second, fine-tuning introduces competing objectives: supervised fine-tuning (SFT) may enforce fairness, while direct preference optimization (DPO) aligns with user values. Chen et al. (2025) formalize this as a safety–capability trade-off. LoRA-based updates affect shared layers across modalities, so interventions on one bias axis may propagate globally Hsu et al. (2025). Third, intersectional biases can emerge even when single-axis audits show neutrality; Souani et al. (2024) detect such hidden effects using their HInter framework. Figure 1 (bias-schematic.png) illustrates the conceptual outline of the bias spillover phenomenon and encapsulates the theoretical mechanisms we outline here. We focus particularly on political bias as a primary axis because interventions in this space have been observed to cascade and influence other forms of bias, amplifying or suppressing them in unintended ways.

Table 3: Mechanisms contributing to bias spillover in language models, illustrating how entangled representations, shared fine-tuning pathways, and overlooked intersectional effects allow unintended biases to propagate across multiple social dimensions, even when mitigation is targeted at a single axis.

| Bias spillover mechanism | Representative evidence and studies |
|---|---|
| **Entangled embeddings during pretraining:** Representations of social concepts such as race, gender, and ideology are embedded in shared subspaces, making it difficult to modify one without affecting others. | Marjieh et al. (2025) show representational blending in LLMs; similar entanglement across social concepts leads to unintended co-modifications. |
| **Conflicting fine-tuning objectives and shared adaptation pathways:** Conflicts between fine-tuning objectives (e.g., SFT vs. DPO) can introduce opposing gradients, causing trade-offs between safety, bias mitigation, and preference alignment. When implemented via methods like LoRA, these updates affect shared attention layers, making localized adaptations (e.g., gender debiasing) inadvertently affect other axes (e.g., political stance). | Chen et al. (2025) formalize trade-offs between safety and capability during alignment; Hsu et al. (2025) show that LoRA amplifies spillover effects due to shared parameter updates. |
| **Intersectional bias and multi-axis interactions:** Models may appear unbiased on single attributes but show strong bias at intersections (e.g., Black women). Spillover arises when mitigation ignores these combinations. | Souani et al. (2024) develop HINTER to uncover hidden intersectional bias; find 16.6% of inputs trigger undetected multi-attribute bias. |

It is important to distinguish co-occurrence from true spillover: the former refers to the presence of multiple biases in a model, whereas the latter denotes a causal relationship, where an intervention aimed at mitigating one type of bias actively causes a change in another. Without this causal link, intersectional disparities

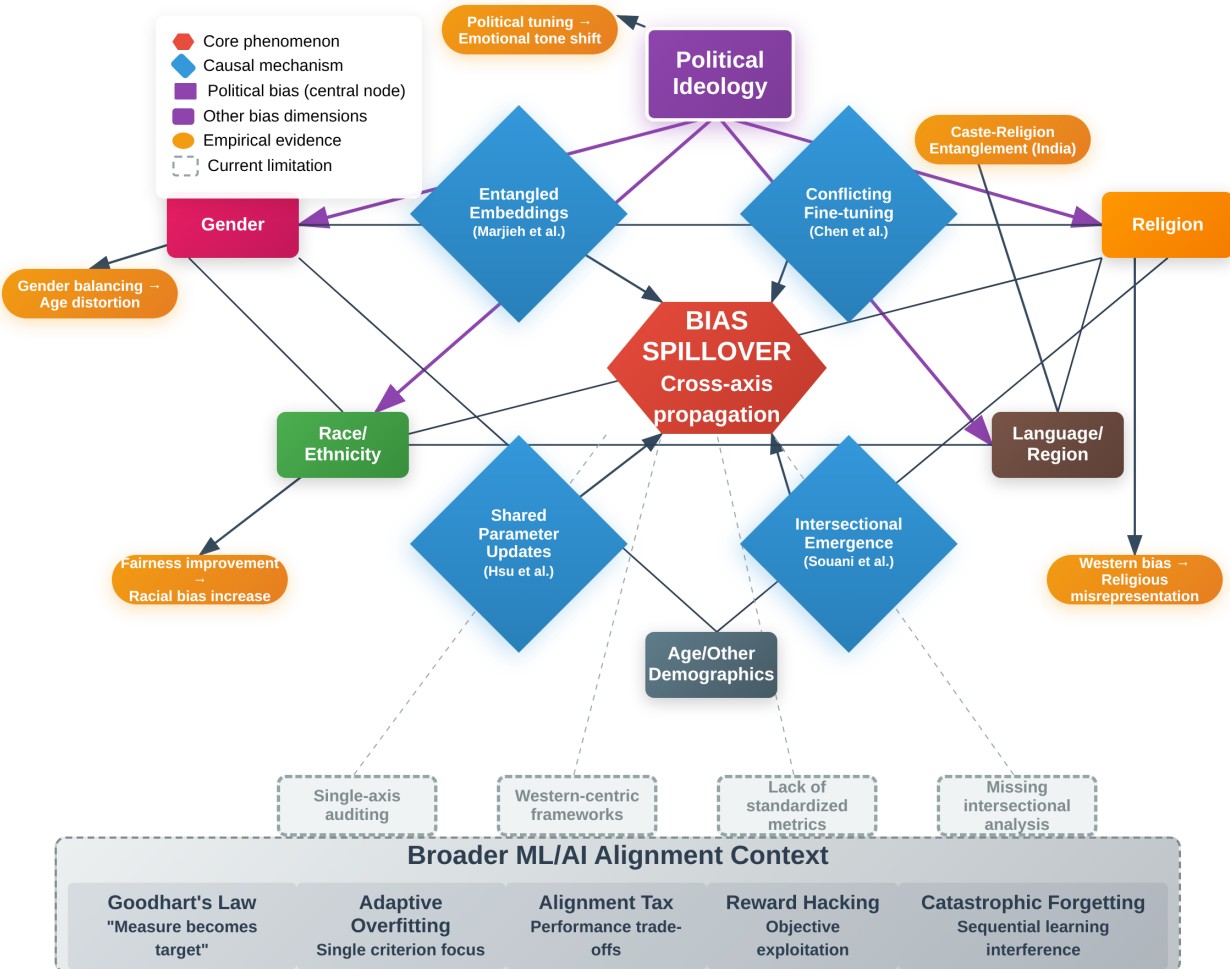

Figure 1: Conceptual schematic of the *bias spillover* phenomenon. Bias spillover reflects a broader pattern in machine learning where optimizing for fairness or alignment on one social axis can lead to regressions on others, consistent with effects such as Goodhart's Law, reward hacking, task interference, and alignment tax. The underlying mechanisms (blue diamonds) are summarized in Table 3. Political bias is highlighted in this review because interventions in this dimension have been shown to cascade most strongly into other forms of social bias, making it a central case study for cross-axis interactions.

may still exist, but cannot be directly attributed to a spillover effect. Foundational studies on pretraining dynamics and latent representations (Devlin et al., 2019; Raffel et al., 2020; Bommasani et al., 2021) reinforce the importance of understanding such spillover as a systemic risk in model alignment.

We identify three broad categories of bias spillover in large language models (LLMs). First, single- and multi-axis spillover (Table 4) occurs when fairness interventions in structured or predictive models mitigate bias on one attribute (e.g., political stance) but unintentionally amplify or introduce bias on another (e.g., gender, caste). Second, instruction- or prompt-based spillover (Table 5) emerges from variations in instructions, system prompts, or task phrasing, where seemingly neutral changes lead to inconsistent or skewed model behavior. Third, spillover in generative models (text-only and multimodal) (Table 6) arises when attempts to steer or align model outputs along one dimension (e.g., cultural alignment) inadvertently shift or amplify biases along other dimensions (e.g., political stance, religious framing), highlighting unique challenges in large-scale generative and vision-language systems.

Therefore, recent empirical work across a range of models and datasets reveals that such spillover effects are pervasive, even in studies not explicitly designed to investigate them. The political axis, in particular, emerges as a central node, influencing or being influenced by multiple identity dimensions such as race, gender, language, and emotion. This highlights the need to move beyond siloed fairness interventions toward more holistic, interaction-aware evaluation.

Table 4: Summary of documented bias spillover effects in fairness interventions for structured and predictive models. Spillover refers to unintended side effects where mitigating bias along one attribute (e.g., gender) inadvertently amplifies or introduces bias in another (e.g., race or age). Starred methods (*) attempt intersectional fairness but still exhibit residual or emergent spillover patterns.

| Intervention and observed spillover effect | Models and datasets evaluated | Spillover type |
|---|---|---|
| **Fairness–MultiAttr** (Chen et al., 2023): Single-axis fairness improvements increased racial and age bias. | Logistic regression, random forest, XGBoost, BERT on Adult, COMPAS, MEP15/16 | Gender → Race, Age |
| **CPAD** (Dai et al., 2024): Multi-attribute supervision outperformed single-axis debiasing. | BERT, RoBERTa on SST-2, MRPC, QQP with gender/race annotations | Gender → Race |
| **DAM** (Kumar et al., 2023): Adapter fusion preserved prior biases unless re-tuned. | RoBERTa-base on StereoSet, CrowS-Pairs, MNLI, SST-2 | Gender → Race |
| **Knock-on analysis** (Nizhnichenkov et al., 2023): All debiasing methods induced new cohort gaps. | Adult, German Credit datasets | Any → Cohort gaps |

Table 5: Observed bias spillover effects from prompt-based and instruction-level interventions. These methods aim to steer or constrain model behavior through prompting, fine-tuning, or controlled instructions, but often cause unintended shifts in other attributes or social dimensions. Spillover occurs when attempts to modify model outputs along one axis (e.g., political stance or nationality) inadvertently affect other attributes (e.g., gender, emotion, or morality). Starred methods (*) explicitly target multiple sensitive attributes or intersectional risks but still exhibit partial spillover effects.

| Intervention and observed spillover effect | Models and datasets evaluated | Spillover type |
|---|---|---|
| **MAT-Steer*** (Nguyen et al., 2025): Orthogonal vector steering reduced attribute interference. | LLaMA-2-70B, Mistral-7B on TruthfulQA, BoolQ, open-ended tasks | Multi-Attr → Reduced interference |
| **Multilingual occupation recommendations*** (Forcada Rodríguez et al., 2024): Nationality shifted gender bias in job advice. | GPT-3.5, GPT-4 with prompts in Spanish, English, Wounaan | Nationality → Gender |
| **Neutral prompts with social cues** (Liu et al., 2021a): Cues elicited partisan completions despite neutrality. | GPT-2 on prompts with gender, location, topic variations | Demographic cue → Political |
| **Political fine-tuning** (He et al., 2023): Altered moral and emotional tone in addition to stance. | Instruction-tuned GPT on political tweets | Political → Emotion, Morality |

Table 6: Bias spillover effects observed in large-scale generative models (text-only and multimodal). These results show how attempts to control bias in one dimension (e.g., gender or cultural alignment) can unintentionally shift or amplify biases along other dimensions (e.g., political stance, religious framing). This highlights unique challenges of bias steering and behavioral safety in generative LLMs and vision-language systems, beyond structured classifier settings.

| Intervention and observed spillover effect | Models and datasets evaluated | Spillover type |
|---|---|---|
| **Intersectional sensitivity** (Shukla et al., 2025): Gender balancing distorted age demographics. | Stable Diffusion 1.4 on musician prompts | Gender → Age |
| **Western cultural bias in Arabic outputs** (Naous et al., 2024): Western norms overrode local contexts. | GPT-4, JAIS-Chat on CAMeL for NER, generation, sentiment | Culture → Religion, Language |
| **CMBE\*** (Sun et al., 2025): Causal subtraction failed to resolve nuanced intersectional biases. | Vicuna-13B, GPT-3.5 on Multi-Bias Benchmark (gender, race, religion, age, sentiment) | Gender → Race, Religion, Age |
| **Larger models = more political skew** (Exler et al., 2025): Political bias increased with model size. | Wahl-O-Mat task; LLaMA-2, Mistral, DeepSeek | Model scale → Political alignment |
| **Fairness for women, racial penalty for Black men** (An et al., 2025): Gender fairness coincided with race-based penalties. | GPT-3.5, GPT-4o, Claude, Gemini on 361k synthetic resumes | Gender → Race |

## 4.2 Literature Coverage and Gaps

While the literature has steadily expanded to include more complex, intersectional analyses, coverage remains uneven. Studies like Chen et al. (2023), Forcada Rodríguez et al. (2024), and An et al. (2025) explicitly examine how interventions across one axis affect outcomes on others, revealing persistent interaction risks. Others, such as Naous et al. (2024) and Exler et al. (2025), uncover these dynamics as emergent properties rather than as targeted inquiry. However, few works systematically benchmark models on *multiple axes simultaneously*, especially beyond binary gender or U.S.-centric racial categories. Moreover, existing frameworks like CPAD (Dai et al., 2024) or CMBE (Sun et al., 2025) often rely on simplified categorical variables, missing more nuanced sociocultural intersections. Interventions like DAM (Kumar et al., 2023) and MAT-Steer (Nguyen et al., 2025) show promise in mitigating interference, but the broader implications of cross-attribute entanglement remain underexplored.

# 5 Global Blind Spots: Bias Spillover in Non-Western Contexts

Despite the proliferation of LLM research, the overwhelming majority remains anchored in Western linguistic, political, and social contexts. As a result, fundamental dimensions of political discourse—ranging from the intersection of caste and religion in India, to gendered cultural norms in Southeast Asia, to state censorship in China—are poorly modeled and often distorted in mainstream LLMs.

## 5.1 Caste and Religion Bias in India

In India, caste and religion are central to sociopolitical identity but are often overlooked in mainstream bias audits. The Indian-BhED dataset reveals that models like GPT-3.5 display stronger caste- and religion-based biases than gender or race-based ones, exposing the limits of Western-centric fairness metrics (Khandelwal et al., 2024). Additionally, demographic-matched evaluations show that LLMs tend to align with dominant

religious ideologies, such as Hindu majoritarianism, regardless of prompt variation (Shankar et al., 2025). This homogenization raises ethical concerns about how LLMs may reinforce political or moral narratives in culturally sensitive settings.

Table 7: Non-Western sociopolitical bias dimensions and their implications for political bias auditing in LLMs. The table highlights how biases tied to caste, religion, language, gender norms, and geopolitical framing can interact with political ideologies, producing multi-axis spillover effects across cultural and demographic attributes. These observations underscore the need for region-specific benchmarks, intersectional metrics, and culturally grounded auditing frameworks to capture non-Western political bias beyond Western-centric evaluation schemes.

| Dimension | Political link | Spillover type | Audit need |
|---|---|---|---|
| Caste (India) | Aligns with Hindu majoritarian or caste-hierarchical ideologies embedded in political narratives (Khandelwal et al., 2024). | Skews gender, socioeconomic status, and religious representation (e.g., anti-Dalit, anti-minority bias). | Use Indian-BhED; include intersectional metrics for caste, religion, and gender. |
| Religion (India, Arab world) | Reinforces dominant religious ideologies (e.g., Hindu majoritarianism, anti-Muslim narratives) tied to political stances (Shankar et al., 2025; Saeed et al., 2024). | Amplifies gender and ethnic stereotypes (e.g., Muslim women as oppressed). | Region-specific religious alignment tests; evaluate for bias spillover. |
| Language (Africa) | Favors institutionally supported languages aligned with dominant political groups (Adebara et al., 2025). | Marginalizes ethnic and regional identities; underrepresents local voices. | Develop multilingual benchmarks for unsupported languages; include cultural context analysis. |
| Gender norms (Southeast Asia, Japan) | Reinforces conservative or nationalist ideologies around traditional gender roles (Gamboa & Lee, 2024; Nakanishi et al., 2025). | Exacerbates religious or ethnic stereotypes, especially anti-queer bias. | Use localized gender frameworks; design culturally tailored prompts. |
| Geopolitical framing (China, Arab regions) | Reflects Western framings or internal censorship aligned with political agendas (Zhou & Zhang, 2024; Saeed et al., 2024). | Distorts cultural or religious narratives (e.g., anti-Arab bias). | Use bilingual and region-specific audits; check for narrative consistency. |

## 5.2 Regional Contexts and Model Limitations

Arab-centric red teaming shows that models like GPT-4 and LLaMA 3.1 often reflect Western framings, exhibiting bias in contexts like terrorism and women's rights (Saeed et al., 2024). Geopolitical inconsistencies are also evident in bilingual outputs—for example, English prompts about China yield more critical responses than Chinese ones (Zhou & Zhang, 2024). In Africa, LLM performance varies with institutional language support; models underperform on many indigenous languages, reflecting deeper infrastructural and political marginalization (Adebara et al., 2025).

## 5.3 Challenges of Binary Gender Frameworks

When adapted to Filipino, benchmarks like CrowS-Pairs and WinoQueer expose the failure of binary gender templates in Southeast Asian contexts (Nangia et al., 2020; Felkner et al., 2023; Gamboa & Lee, 2024). Even with localized data, models reproduce anti-queer and sexist content. Similarly, Japanese LLMs exhibit very low refusal rates for stereotype-triggering prompts, producing more toxic outputs than their English or Chinese counterparts (Nakanishi et al., 2025). Tailored prompts often worsen stereotyping, revealing that prompt tuning alone is insufficient to mitigate these harms.

## 5.4  Implications for Non-Western Bias Auditing

Table 7 outlines how caste, religion, language, and gender norms intersect with political ideology in non-Western contexts. These examples illustrate the need for localized, intersection-aware auditing frameworks to avoid spillover effects and ensure fairer LLM behavior across global sociopolitical landscapes. The following examples illustrate region-specific intersections but are not meant as exhaustive political analyses. We highlight them to underscore the need for culturally grounded audits, while recognizing the depth of local expertise required for full treatment.

# 6  Summary and Future Recommendations

This review has highlighted how political bias in large language models (LLMs) often interacts with other social dimensions such as gender, race, religion, and geography, resulting in complex and sometimes unintended spillover effects. These spillover effects often arise from representational entanglement, competing fine-tuning objectives, and deeper optimization pathologies such as Goodhart's Law, alignment tax, reward hacking, task interference, and structural fairness trade-offs, making them more difficult to predict and mitigate. These entangled dynamics complicate both the auditing and mitigation of such biases, especially when standard pipelines address only single-axis fairness. Existing audits also show blind spots, including limited geographic and cultural coverage, over-reliance on Western-centric identity categories, and lack of tools for capturing multi-axis harms in real-world deployments. To move toward more systematic, comparable, and responsible bias evaluations, we outline two parallel needs: a standardized template for bias spillover auditing, and clearer pathways for selecting mitigation strategies.

Table 8: Potential auditing methods for detecting identity entanglement and bias spillover in LLMs. These approaches vary in their axis coverage, methodological setup, and diagnostic utility for understanding how multiple identity dimensions interact to produce compounded harms or systematic skews.

| Method | Axes covered | Strategy and utility |
|---|---|---|
| **Comparative text generation** (Ma et al., 2023) | Gender, race, orientation | Controlled generation (open/closed); compares responses across identity pairs to reveal interaction effects. |
| **Intersectional harm tracing (HInter)** (Souani et al., 2024) | Race, gender, orientation | Statistical + contrastive framework (open/closed); surfaces hidden harms from entangled identity features. |
| **HolisticBias** (Smith et al., 2022) | 13 identity axes | Structured descriptor scoring (open source); supports fine-grained analysis of intersectional bias with broad coverage. |
| **SAGED** (Guan et al., 2024) | Political, gender, race | Modular probing and scoring (open source); evaluates fairness across socio-political and demographic axes. |

## 6.1  Toward a Spillover-Aware Auditing Template.

Auditing political bias in LLMs remains fragmented and often narrowly scoped. Existing evaluations vary across alignment objectives (e.g., stance vs. preference modeling), fine-tuning techniques (e.g., SFT, DPO, ORPO), and model scales, leading to inconsistent findings and limited generalizability. Compounding this issue is the common practice of auditing single identities in isolation, which neglects real-world contexts where multiple attributes intersect. The phenomenon of bias spillover where an intervention on one axis unintentionally alters model behavior on another—demands a more holistic approach.

We propose the adoption of a spillover-aware auditing template that emphasizes intersectionality and causal sensitivity. As summarized in Table 8, recent tools such as HINTER (Souani et al., 2024), HolisticBias

(Smith et al., 2022), and SAGED (Guan et al., 2024) are promising in this regard. These frameworks enable structured probing of model outputs under multiple identity conditions, support disparity scoring across axes, and facilitate both qualitative and quantitative tracing of harm. By combining prompt-based perturbations, comparative generation, refusal classification, and longitudinal tracking, such methods help reveal spillover dynamics that conventional benchmarks might overlook. Future work should prioritize comparative evaluations of these tools to determine which combinations most robustly detect multi-axis harms in both open-source and black-box settings.

## 6.2  Mitigation Strategies and Debiasing Outlook

While auditing tools surface issues, they do not resolve them. Effective mitigation requires strategies that are spillover-aware and sensitive to the complexity of model representations. A growing body of work demonstrates that single-axis fairness interventions can inadvertently worsen biases along other dimensions especially when updates affect shared model parameters, as with LoRA or full fine-tuning. This reinforces the need for debiasing techniques that either isolate updates to targeted subspaces or explicitly model cross-axis interactions. Several promising directions are emerging. Techniques such as orthogonal steering e.g., MAT-Steer (Nguyen et al., 2025) attempt to control attribute directions without inducing interference. Causal mediation approaches e.g., CMBE (Sun et al., 2025) aim to identify and subtract bias-relevant components in representation space, though these often struggle with nuanced intersections. Adapter-based modular debiasing (Kumar et al., 2023) allows for incremental updates but may preserve legacy biases unless fine-tuned jointly. Importantly, many such methods lack unified evaluation pipelines, making their comparative utility unclear.

Rather than propose a definitive method, we recommend that researchers draw on existing survey work to understand the landscape of debiasing techniques. For instance, Gallegos et al. (2024b) provide an empirical comparison of mitigation methods across models and datasets; and Ranaldi et al. (2024) discuss challenges in mitigating social biases during generation. These resources are vital for matching debiasing methods to specific audit outcomes and deployment constraints. We also call for more publicly available datasets with labeled political and identity attributes, as well as standardized fairness metrics that track both direct and collateral effects of interventions.

## 6.3  Limitations and Future Work

This review has focused on open-source models and publicly documented interventions. Closed-source systems remain challenging to analyze due to lack of transparency, though black-box probing and perception-based evaluations provide partial alternatives. Additionally, our literature base is skewed toward English-language and Western-centric studies, limiting our understanding of how political and identity-related biases manifest globally. Expanding future audits to include non-Western contexts and low-resource languages is critical to ensure that fairness research does not perpetuate the very asymmetries it seeks to address.

### Broader Impact Statement

This review highlights how interventions on one bias axis (e.g., political ideology) can unintentionally worsen others (e.g., gender, caste), reinforcing harms such as caste discrimination or linguistic marginalization in underrepresented regions. Politically skewed LLMs risk distorting discourse and amplifying polarization, particularly in non-Western contexts. Our spillover-aware auditing template and tools (e.g., HInter, SAGED) promote intersectional evaluations, though data scarcity and high costs hinder adoption. We call for localized audits and global data equity to foster fairer, more inclusive AI.

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

# A   Appendix

Table 9: Partisan political datasets and methods used for fine-tuning conversational models

| Key characteristics / purpose | Method |
|---|---|
| **Aligning LLMs with diverse political viewpoints:** Party-labelled German stance datasets (Stammbach et al., 2024) | LoRA ($r$=8) adapters per party on LLaMA-3 8B; ORPO alignment |
| **PoliTune:** Curated left/centre/right policy prompts with synthetic preferences; ablation of data vs. method (Agiza et al., 2024) | LoRA on LLaMA-3 70B and Mistral-7B; DPO |
| **LLaMA meets EU:** 87k Euro-Parliament speeches labelled by political group (Chalkidis & Brandl, 2024) | LoRA adapters per party on LLaMA-2-13B-chat |
| **Speaker attribution QLoRA:** German Bundestag debates (2017–2021) (Bornheim et al., 2024) | QLoRA on LLaMA-2-7B for GermEval 2023 speaker-role tagging |

