# OpenReview forum: "Bias Spillover in Language Models: A Review of Political Alignment, Regional Fragility, and Multi-Axis Risks"
_TMLR — Rejected by TMLR_

### Review · Reviewer_A74u · 2025-07-10

**Summary Of Contributions:**

LLMs present particular risks of social harm to their marginalized group users; inheriting and reproducing stereotypes, ideological distortions, and exclusionary language. These harms often reflect what is broadly referred to as social bias - the systematic disparity in model behavior towards different demographic or ideological groups, arising from historical and structural asymmetries.

In this review paper, the authors provide definitions and typologies of political ideology, enumerate methods for political bias auditing, include a brief section on US/EU political differences and how these cultural and political distinctions lead to non-synchronous behavior in LLMs-as-political-analysts, and spend the remainder of the paper defining and analyzing the phenomenon of "spillover bias", a phenomenon where mitigating bias along one social axis (e.g., political ideology) unintentionally alters model behavior on another (e.g., gender or race).

**Audience:**

Yes

**Claims And Evidence:**

Yes

**Requested Changes:**

As this work is a review, I think it is particularly important for the authors to more fully document the landscape in which their study resides.

1. Rather than introducing "spillover bias" as a wholly novel concept, as the authors currently do, the authors should make an attempt to contextualize and situate their work more fully within the long history of studying overfitting in machine learning. Adaptive overfitting, for example, (https://arxiv.org/abs/1506.02629) describes a general phenomenon quite similar to the one documented by the authors of this work in one particular domain (social bias). Similar phenomena are studied under the heading of distributional robustness (https://proceedings.neurips.cc/paper/2020/file/d8330f857a17c53d217014ee776bfd50-Paper.pdf). Seminal works on LM alignment (https://arxiv.org/abs/2203.02155) popularize the concept of an "alignment tax", which, applied to the subdomain of fairness, neatly describes this phenomenon. Other highly related terms the authors should consider when gathering related work: "Goodhart's Law", "Reward hacking", "task interference", and "catastrophic forgetting".

2. The authors also fail to reference the long list of existing impossibility results in fairness, mostly with respect to the multiple conflicting mathematical definitions of fairness (https://arxiv.org/abs/1609.05807); a web survey here (https://cacm.acm.org/research/the-impossibility-of-fairness/) provides more detail.

3. The aesthetics of the paper could stand to be improved; the paper does not contain a single figure, and all of the tables are densely packed with text; some of the tables are extremely large, and the captions for tables are minimal, providing very limited utility.

**Strengths And Weaknesses:**

STRENGTHS:

This useful and timely review can help TMLR readers better understand the evolving state of fairness research, which has begun to question the utility of methods which overfit one particular aspect of fairness while perhaps degrading performance on others. The work is well-written, and the careful definitions of key terms are much appreciated -- such definitions are often lacking in works of this kind. Although I am not an expert in recent fairness methods, the authors appear to have done some careful survey work and have collected many such methods and provided useful comparisons; the tables they put together help highlight just how comprehensive the challenge of "spillover bias" really is. All of these virtues lead to a paper with a clear and well-reasoned perspective.

WEAKNESSES: See below.

---

### Review · Reviewer_XcKL · 2025-07-11

**Summary Of Contributions:**

This paper reviews the phenomenon of bias spillover in LLMs, where mitigating one type of bias (e.g., gender) may unintentionally introduce another (e.g., political). The authors outline potential mechanisms underlying this issue, including entangled representations during pretraining, conflicting objectives in fine-tuning (e.g., SFT vs. DPO), global effects from highly shared parameter updates such as LoRA, and the emergence of intersectional biases. Drawing on prior work, the paper aims to provide a unified conceptual framework to guide future research in bias mitigation.

**Audience:**

Yes

**Broader Impact Concerns:**

The authors have already outlined potential broader impacts of their work within the paper.

**Claims And Evidence:**

No

**Requested Changes:**

- While the paper is structured as a review, many of its key claims—particularly the notion of bias spillover arising from neutral supervision and the propagation of regional political bias—are not supported by concrete empirical evidence. The authors are encouraged to either (a) include systematic empirical results or meta-analyses of existing studies, or (b) clearly frame these points as hypotheses or observations informed by real-world behaviors of LLMs.

-  In addition, the authors propose several mitigation strategies, which should be empirically tested in LLMs to assess whether they can effectively reduce bias and ensure their practical viability.

**Strengths And Weaknesses:**

**Strengths**

- The paper introduces the novel concept of bias spillover, highlighting how mitigating one bias dimension (e.g., gender) may inadvertently introduce others (e.g., political alignment).

- It raises an important observation that no single PEFT method maintains optimal fairness across all political alignment tasks, offering practical insight into the limitations of current fine-tuning techniques.

- Through examples from the U.S. and Europe, the paper discusses the difficulty of designing political alignment and auditing strategies under diverse ideological frameworks, which is a timely and underexplored topic.

**Weaknesses**

- While framed as a review, many of the paper’s central claims—such as bias spillover from neutral supervision or the transmission of regional political bias—lack direct empirical validation or systematic meta-analysis of existing studies. As a result, the arguments feel speculative and leave readers navigating a landscape of concepts without concrete evidence.

- Tthe discussion of SFT-DPO conflicts (as competing objectives) and LoRA-induced global effects are listed as separate causes of bias spillover. However, LoRA is merely a method to implement SFT or DPO. The separation of these points may reflect conceptual overlap and could benefit from clearer consolidation or distinction.

---

### Review · Reviewer_fraS · 2025-07-25

**Summary Of Contributions:**

The premise of the paper is a survey of "bias spillover", which occurs when debiasing on one attribute introduces new biases on other attributes. The paper does not introduce any new evidence of bias spillover, but it reviews prior findings. The paper also highlights other problems with bias auditing and debiasing, such as challenges in transferring across regions, which do not have an obvious direct relationship to the bias spillover concept.

**Audience:**

No

**Broader Impact Concerns:**

No major concerns, but as suggested above I think it's risky to lump "political debiasing" together with the more fundamentally egalitarian intervention of debiasing on protected characteristics like race and gender. Neutrality and moderation are political positions too, so it's not clear what it would even mean for a language model to be debiased in this sense.

**Claims And Evidence:**

No

**Requested Changes:**

Unfortunately I think the paper is pretty far from a state where I would recommend acceptance. Focusing more narrowly on bias spillover and more systematically characterizing the resulting issues (and ideally, their mechanistic causes) would be the right path forward.

**Strengths And Weaknesses:**

On the positive side, the paper targets an important issue in the impact of alignment interventions such as debiasing on other aspects of LLM behavior. In general, the potential side-effects of late-stage alignment techniques like RLHF is an open question that is both technically deep and practically relevant, and is particularly salient in the context of challenging alignment concepts like bias. There seem to be increasingly direct attempts to control the political orientation of some language models (e.g. https://www.pbs.org/newshour/politics/why-does-the-ai-powered-chatbot-grok-post-false-offensive-things-on-x), and it would be good to know whether such interventions are possible without affecting other seemingly unrelated characteristics of the LLM.

But while the concept of bias spillover is potentially insightful, the paper lacks focus, bringing in a number of other tangentially-related issues, which, while valid, do not appear to be examples of spillover: overemphasis of the US political context, challenges in transferring bias audits to multi-polar political systems, lack of debiasing for caste, etc. I also found that the subsection headings in section 5 often make assertions that don't seem to be supported by the text: for example, I fail to see what part of section 5.1 supports the assertion that "caste and religion are inseparable from politics", or what part of 5.3 shows that "western gender templates are misapplied."  These claims may well be true, but the text of the paper doesn't support it.

The introduction to the paper focuses heavily on political bias, but I'm not sure whether it's appropriate to think about political orientation in the same way as gender and race bias. In the case of the latter, it should be possible to get a rough handle on the intended behavior from the starting position that AI technology should people equally regardless of protected characteristics like race and gender. For political orientation, it's less clear what behavior is desired. Should the LLM consider every possible political position to be equally valid? Or is the idea that on aggregate, the model should occupy the exact center of a region's political spectrum? In any case, despite the focus on political bias in the early sections of the paper, table 5 doesn't seem to show any examples of spillover from political debiasing.

As a more minor issue, the writing and typesetting (esp. references) are pretty sloppy and could use another revision.

---

### Decision · Action_Editor_u1Qw · 2025-08-26

**Recommendation:** Reject

**Additional Comments:**

After considering the reviews and the author responses, I conclude that the submission does not meet TMLR’s criterion that “the claims made in the submission are supported by accurate, convincing, and clear evidence.” While the topic is timely and likely of interest to parts of the TMLR audience, the evidentiary support for the paper’s central assertions (including the technique categorization and suggested future direction) remains insufficient.

**Audience:**

Yes

**Audience Explanation:**

Exploring bias spillover in language models is an important area of study for their deployment across critical domains.

**Claims And Evidence:**

No

**Claims Explanation:**

In a literature review, the proposed categorization/typology (of “bias spillover” and its mechanisms) and the recommendations for future work function as the paper’s core claims -- they are the value the review adds beyond listing prior studies. These claims must be grounded in a transparent, accurate synthesis of prior evidence. From this perspective, the current version does meet this criterion. For example, one key limitation lies in **definition–evidence mismatch**. The paper defines “bias spillover” clearly, but then intermixes it with broader fairness topics and examples that do not meet the stated definition in Sections 3 & 4.